IdentPMP: identification of moonlighting proteins in plants using sequence-based learning models

Liu Xinyi
Shen Yueyue
Zhang Youhua
Liu Fei
Ma Zhiyu
Yue Zhenyu zhenyuyue@ahau.edu.cn
Yue Yi yyyue@ahau.edu.cn
School of Information and Computer, Anhui Provincial Engineering Laboratory for Beidou Precision Agriculture Information, Anhui Agricultural University , Hefei , Anhui , China
Arora Gunjan
Electronic publication date: 2021 Aug 6
Publication date: 2021
Volume: 9
Electronic Location ID: e11900
Received 2020 Nov 17; Accepted 2021 Jul 13
Copyright: ©2021 Liu et al.
Copyright year: 2021
Copyright holder: Liu et al.
License: This is an open access article distributed under the terms of the Creative Commons Attribution License, which permits unrestricted use, distribution, reproduction and adaptation in any medium and for any purpose provided that it is properly attributed. For attribution, the original author(s), title, publication source (PeerJ) and either DOI or URL of the article must be cited.
License URL: https://creativecommons.org/licenses/by/4.0/

Keywords: Prediction tool, Plant moonlighting protein, eXtreme gradient boosting, Benchmark data set

Funding: Natural Science Young Foundation of Anhui 2008085QF293 “Three Renewal and One Creation” Innovation Platform Fund-Anhui Provincial Engineering Laboratory for Beidou Precision Agriculture lnformation (Anhui Development and Reform Innovation [2020]555 Natural Science Young Foundation of Anhui Agricultural University (2019zd12) Introduction, Stabilization of Talent Project of Anhui Agricultural University yj2019-32 Graduate Innovation Fund of Anhui Agricultural University 2021yjs-53 This work was supported by the grants from the Natural Science Young Foundation of Anhui (2008085QF293), the 2020 “Three Renewal and One Creation” Innovation Platform Fund-Anhui Provincial Engineering Laboratory for Beidou Precision Agriculture lnformation (Anhui Development and Reform Innovation [2020]555), the Natural Science Young Foundation of Anhui Agricultural University (2019zd12), and the Introduction, Stabilization of Talent Project of Anhui Agricultural University (yj2019-32) and the Graduate Innovation Fund of Anhui Agricultural University (2021yjs-53). The funders had no role in study design, data collection and analysis, decision to publish, or preparation of the manuscript.

==============================
Background

A moonlighting protein refers to a protein that can perform two or more functions. Since the current moonlighting protein prediction tools mainly focus on the proteins in animals and microorganisms, and there are differences in the cells and proteins between animals and plants, these may cause the existing tools to predict plant moonlighting proteins inaccurately. Hence, the availability of a benchmark data set and a prediction tool specific for plant moonlighting protein are necessary.

Methods

This study used some protein feature classes from the data set constructed in house to develop a web-based prediction tool. In the beginning, we built a data set about plant protein and reduced redundant sequences. We then performed feature selection, feature normalization and feature dimensionality reduction on the training data. Next, machine learning methods for preliminary modeling were used to select feature classes that performed best in plant moonlighting protein prediction. This selected feature was incorporated into the final plant protein prediction tool. After that, we compared five machine learning methods and used grid searching to optimize parameters, and the most suitable method was chosen as the final model.

Results

The prediction results indicated that the eXtreme Gradient Boosting (XGBoost) performed best, which was used as the algorithm to construct the prediction tool, called IdentPMP (Identification of Plant Moonlighting Proteins). The results of the independent test set shows that the area under the precision-recall curve (AUPRC) and the area under the receiver operating characteristic curve (AUC) of IdentPMP is 0.43 and 0.68, which are 19.44% (0.43 vs. 0.36) and 13.33% (0.68 vs. 0.60) higher than state-of-the-art non-plant specific methods, respectively. This further demonstrated that a benchmark data set and a plant-specific prediction tool was required for plant moonlighting protein studies. Finally, we implemented the tool into a web version, and users can use it freely through the URL: http://identpmp.aielab.net/.

Introduction

The continuous accumulation of technology and bioinformatics data in the post-genome era has brought new opportunities and challenges to bioinformatics research. The phenomenon of protein multi functionality in proteomics has also attracted high attention. As more and more proteins are studied in-depth, they are found to have two or more different functions. The idea of one-to-one correspondence between protein and function has been gradually overturned by moonlighting proteins that can perform two or more functions, that usually refers to a single polypeptide chain rather than the product of gene fusion mutation (Jeffery, 1999; Bo et al., 2019). Moonlighting proteins can perform multiple functions simultaneously or alternately due to triggering certain factors. It has been suggested that moonlighting proteins have a combined effect on cell activities, acting as a switch between certain functions, and also regulating the cell microenvironment (Zanzoni, Ribeiro & Brun, 2019). Moonlighting proteins exist in various organisms, and they undertake many vital functions such as regulation, transcription, and catalysis (Irving et al., 2012; Irving, Cahill & Chris, 2018; Świezawska et al., 2018). These potential functions save costs and increase efficiency by producing fewer proteins from compressed genomes and completing more functions (Luís et al., 2017). It is not easy for moonlighting proteins to be discovered by biological experiments, because certain functions of some proteins are prominent or particularly important that other potential functions are easily overlooked. Therefore, effective methods are needed to summarize or identify such proteins.

Herein, we briefly reviewed the moonlighting proteins databases and some methods for identifying moonlighting proteins. In 2014, Jeffery’s laboratory constructed a database MoonProt, which provides a searchable, web-based database of known moonlighting proteins, containing more than 200 experimentally verified moonlighting proteins (Mathew et al., 2015). Until 2018, the number of proteins in the MoonProt 2.0 database increased to 370, and dozens of protein annotations with additional functions or updated information were modified (Chen et al., 2018). In the same year, Spanish researchers updated the MultitaskProtDB moonlighting protein database, compiled a series of multi-functional moonlighting proteins found in the literature. The existing MultitaskProtDB-II data increased to 694 (Luís et al., 2017). It was not until 2019 that the first comprehensive plant moonlighting protein database PlantMP (Bo et al., 2019) was constructed, which has brought a lot of inspiration to our research.

Daisuke Kihara’s team has developed three tools to predict moonlighting proteins based on different protein types. Initially, they developed a program to identify moonlighting proteins by clustering GO terms (Khan et al., 2014). DextMP is a tool that used the functional descriptions of the proteins to mining moonlighting proteins from literatures (Jain, Gali & Kihara, 2018). MPFit (Khan & Kihara, 2016) used multiple features from GO and multi-omics data, which is a state-of-the-art non-plant specific methods.

Current prediction tools are mainly to identify the moonlighting proteins of animals, microorganisms and other organisms, but not constructed for plants. The plant cell structure is quite different from animal and microbial. Most of the space in plant cells is occupied by large vacuoles in the center, the only remaining space has higher requirements for regulating cell biochemical reactions such as signal ligands, molecules, and cofactors (Wong et al., 2018). The identification method of plant moonlighting protein is different from other proteins. Moreover, the experimental results on plant proteins also prove that the existing tools are not accurate enough. Due to the particularity of plant cell structure, a suitable approach is needed to discover the moonlighting protein of plants. In this context, studying a machine-learning-based plant moonlighting protein prediction tool can better serve the work of plant science and proteomics. In this article we proposed a new moonlighting protein prediction tool, IdentPMP (Identification of Plant Moonlighting Proteins), which used a benchmark data set from multiple different plant species to train the model. We extracted multiple protein feature classes, and selected the features that perform best to construct the prediction model. We expect that the plant proteins could be classified more accurately when introduced into the newly constructed IdentPMP tool.

Materials & Methods

The entire construction process of IdentPMP includes data preparation, feature engineering, construction and prediction models evaluation. The detailed experimental procedure is described next.

Data preparation

PlantMP (Bo et al., 2019) is a relatively comprehensive moonlighting proteins Database of a plant. It contains 147 proteins involving 13 plant species, with Arabidopsis being the most abundant. The majority of data in PlantMP is extracted from PubMed articles (Bo et al., 2019). Besides, to expand the data set, we manually screened some moonlighting protein databases. 40 plant moonlighting proteins were found, including five in the MoonProt 2.0 database (Chen et al., 2018) and 35 in MultitaskProtDB-II (Luís et al., 2017). After removing some proteins that are not recognized by UniProt (Apweiler, 2004) and are unable to obtain sequences, the remaining 152 were used as positive samples in the data set. Among them, 40 proteins from MoonProt 2.0 and MultitaskProtDB-II were used as positive data in the independent test set, and 112 proteins in PlantMP were used as positive data in the training set.

In order to obtain sufficient negative samples for training models, some single-function plant proteins were selected through the following steps. First, we collected 60,000 proteins from eight species of Arabidopsis, Hordeum, Pisum sativum, Oryza sativa, Nicotiana, Pea, Moss and Zea mays on UniProt, to avoid the redundancy of a single species. Second, we used the DAVID tool (Jiao et al., 2012) to acquire GO terms annotation of 60,000 plant proteins, and selected those proteins with at least three GO terms in the Biological Process (BP). Next, we used the GOSemSim package (Wang, 2010) to calculate the semantic similarity of several GO terms for each protein. If the semantic similarity score of several GO terms for one protein is between 0.6 and 1, indicating that the GO terms of this protein have similar meanings, and we regarded it as a plant non-moonlighting protein. Through this process, 306 negative samples (188 Arabidopsis, 58 Oryza sativa, 43 Zea mays, 4 Hordeum vulgare, 9 Nicotiana, 2 Pea, 2 Moss) were selected. For each species, we selected two-thirds of the data as the training data, and round-down (round-up) the indivisible data to the nearest integer. 

Next, to reduce effect of redundant sequences in the data set, we apply a technique to decrease the redundant sequences by the Cd-hit (Li & Godzik, 2006). Cd-hit is a practical tool for clustering biological sequences to reduce sequence redundancy, and increase the significance of other sequence. We use the command, ’cd-hit’ to process the training set with a threshold of 0.7, and use the command ’cd-hit-2d’ to decrease the sequence redundancy between the test set and the training set. The result shows that there are 103 and 35 for the training and test set in positive samples, respectively. In negative samples, there are 155 and 90 for the training and test set, respectively. These data constitute the benchmark data set, and the description of the data set sources can be seen in Fig. 1A.

Figure 1 The flowchart of IdentPMP development.

(A) Data preparation. The composition and source of training data set and independent test data set. (B) Feature engineering. We used iLearn to generate feature classes, perform pre-processing, and use each feature class to construct a classifier to select the best feature. (C) Model training. Five algorithms are used, including Random Forest (RF), Support Vector Machine (SVM), Extreme Gradient Boosting (XGBoost), Decision Tree (DT) and K-Nearest Neighbour (KNN).

Feature engineering

To construct an initial feature pool of plant proteins, obtaining the protein sequence in fasta format on UniProt is preliminary work. Next, we used a tool for protein feature extraction, iLearn is an ensemble platform for feature engineering analysis modeling of DNA, RNA and protein sequence data (Chen et al., 2019). We extracted 16 of feature classes using prepared sequences by iLearn, feature refers to a feature value used to train the model, feature class refers to a group of some features. The 16 feature classes including TPC (Tri-Peptide Composition), CKSAAP (Composition of k-spaced Amino Acid Pairs), CKSAAGP (Composition of k-Spaced Amino Acid Group Pairs), KSCTriad (k-Spaced Conjoint Triad), DDE (Dipeptide Deviation from Expected Mean), CTDD (Distribution), Moran (Moran correlation), GTPC (Grouped Tri-Peptide Composition), Geary (Geary correlation), NMBroto (Normalized Moreau-Broto Autocorrelation), QSOrder (Quasi-sequence-order), CTDC (Composition), CTDT (Transition), PAAC (Pseudo-Amino Acid Composition), SOCNumber (Sequence-Order-Coupling Number), APAAC (Amphiphilic Pseudo-Amino Acid Composition).

TPC is the feature class with the most features. The TPC includes 8000 features, defined as: fr,s,t=NrstN−2′r,s,t∈A,C,D,…,Y

where Nrst is the number of tripeptides represented by amino acid types r, s and t (Bhasin & Raghava, 2004). Tripeptides are composed of three amino acids linked by a peptide bond, its properties and functions are determined by the presence of amino acids and the order in which they appear. The brief introduction of other feature classes can be seen in Table S1.

Subsequently, each type of feature data in feature pool is processed separately. Here, we briefly describe the three major steps in the following (Fig. 1B). (1) For the first step, a well-known feature selection technique, Information Gain (IG), was adopted. IG measures the amount of information in bits with respect to the class prediction (Chen et al., 2010; Chyh-Ming, Wei-Chang & Chung-Yi, 2016). The predictive accuracy of the classifier solely depends on the information gained during the training process. We use an information entropy greater than 0.05 as a threshold to determine the number of feature selections for each feature class. There are some feature classes in which the information entropy of all features is less than 0,05. Although these feature classes are less effective in constructing models, we did not discard it. For these feature classes, we selected the top 80% of the features with information entropy. The feature dimension extracted by all feature classes can be queried in Table S1 (we have added experiments on feature classes with information entropy less than 0.05. The results of feature selection 70%, 80%, and 90% are shown in Tables S2–S4). (2) Secondly, the feature class are in different orders of magnitude. In order to solve the comparability between the characteristic indexes, make the process of the optimal solution smooth, and improve the calculation accuracy, this experiment used the minimax normalization method (Shalabi, Shaaban & Kasasbeh, 2006) so that the indexes are in the same order of magnitude, which is convenient for comprehensive comparison. (3) The third step of data processing is dimension reduction. The method we chosen is PCA (Principal Component Analysis), which is used to decompose a multivariate dataset in a set of successive orthogonal components that explain a maximum amount of the variance, and each feature class is reduced to 10 dimensions (Pearson, 1901).

After dimension reduction in the previous step, the 16 classifiers were constructed using each of the 16 feature classes by Support Vector Machine (SVM) (Furey et al., 2000). The selection of kernel and parameters of SVM has an important effect on the performance of classifier, and we use grid-search algorithm to choose the optimal parameters of SVM. Then the performances of classifiers have been evaluated by 5-fold cross-validation, all training sets are randomly divided into five equally sized subsets. The cross-validation process is performed five times, and in each validation, a subset is selected as the test set and the remaining four as the training set. After constructing the 16 classifiers, we analyze the performance of each classifier and rank it. The feature classes used by the best performing classifier will be used in our prediction tool.

Construction and prediction models evaluation

It is important to choose a suitable classification prediction algorithm. For this purpose, we used four algorithms, Extreme Gradient Boosting (XGBoost), Support Vector Machine (SVM), Random Forest (RF), Decision Tree (DT) and K-Nearest Neighbour (KNN) to build the prediction models for plant moonlighting proteins. The selection of the model algorithm is shown in the flowchart Fig. 1C.

SVM is a powerful machine learning algorithm for binary classification (Furey et al., 2000). It aims to accurately classify samples by generating optimal hyperplanes based on the feature dimensionality of the training data (Vapnik, 1999). RF is well-established and widely employed algorithm, which has been applied for many bioinformatics applications (Jia et al., 2016). It is essentially an ensemble of a number of decision trees, built on N random subsets of the training data, and the average prediction performance is usually reported in order to avoid over-fitting (Breiman, 2001). XGBoost is an ensemble algorithm, which is scalable machine learning system for tree boosting, and based on the integration of classification and regression trees (Chen & Guestrin, 2016). DT apply a tree-shaped decision model and only contains conditional control statements, which is a common and effective classification algorithm. KNN algorithm is another commonly employed unsupervised algorithm that clusters samples by calculating their similarities (Cai et al., 2012). This method is easy to understand and implement.

Among the existing related tools, MPfit is a suitable choice for comparison with our prediction tool. We compared the performance of IdentPMP and MPfit on the independent test set. To compare different machine learning methods for constructing prediction models from the training set model, we used six commonly used metrics, including AUPRC (area under the precision–recall curve), AUC (area under the receiver operating characteristic curve), MCC (Mathews correlation coefficient), F1-score, sensitivity and specificity. Among them, sensitivity is the true positive rate, which refers to the proportion of samples that are actually positive, specificity is the true negative rate, which refers to the proportion of samples that are actually negative. AUPRC, AUC are commonly used comprehensive evaluation metrics. IdentPMP is aim to predict plant moonlighting proteins, which are positive samples. Compared with AUC, AUPRC can better evaluate a model’s ability to correctly predict and select positive samples, and is a more suitable metric for evaluating IdentPMP. AUPRC has higher requirements for positive samples in its evaluation performance, and when the prediction of positive samples is incorrect, the penalty will increase. This is more in line with our expectation to identify moonlighting protein. Therefore, we regard AUPRC as the primary metric in this experiment.

Results and Discussion

Selection and analysis of feature class

The 16 feature classes extracted through iLearn, was mentioned in the previous section to construct classifiers, respectively using SVM algorithms. We used AUPRC as the primary metric, it can take a relatively comprehensive evaluation of the model. AUC is also regarded as an essential metrics. They do not need to set any specific threshold when evaluating model performance. Besides, the threshold-based metrics are calculated as minor criteria, i.e., Sensitivity, Specificity, MCC, F1-score. From Table S3, we can see that TPC performs much better than other feature classes. . Anishetty, Pennathur & Anishetty (2002) demonstrated that the tripeptide might be used to predict plausible structures for oligopeptides and de novo protein design. Tripeptide motifs represent potentially crucial for the design of small-molecule biological modulators.

We also integrated TPC with other feature classes to construct models, but the results were not as good as using TPC alone. That shows that integrating these feature classes do not necessarily improve performance. The possible reason is that the feature classes extracted by iLearn are sequence-based, and TPC already contains useful features related to the sequence, including redundant information with other features. Based on the above discussion, we adopted TPC to predict plant moonlighting protein and then built the predictive model in the following study.

Comparison of different classification algorithms

There are various differences in each classification algorithms. Using different classification algorithms to build models will affect the performance of prediction tools. We compared five commonly used algorithms (SVM, RF, XGBoost, DT and KNN) to analyze the impact of different algorithms on performance. We used the TPC feature class to train the classifier with five algorithms, respectively, and used the adaptive optimization method of grid search to optimize the learning model.

The 5-fold cross validation and training set results of five classifiers were shown in Table 1. As we can see from the table, the XGBoost algorithm has the best performance on AUPRC, AUC, Sensitivity, MCC and F1-score. To show the performance of these metrics more clearly, we plotted AUPRC and AUC curves of different classifiers in Fig. 2. Then we can observe that AUPRC (AUPRC =0.85) of XGBoost is better than other algorithms, the AUC results of XGBoost and SVM are similar at 0.87. In conclusion, XGBoost provides stronger identification capability than the other algorithms and is more appropriate for handling with the experiment of distinguishing plant moonlighting proteins from non-moonlighting proteins. Moreover, the results of the five algorithms on the independent test set are shown in Table S5, it can be seen that XGBoost performs better in the comprehensive metric with 0.5 as the threshold. We checked the predicted result and analyzed some of the proteins that were predicted to be positive samples. The protein (UniProt ID Q9ZVR7) is predicted as a moonlighting protein by IdentPMP. A recent article analyzed this protein and confirmed that it is a moonlighting protein (Turek & Irving, 2021). Q38970, which is predicted to be a moonlighting protein, and its different functions have also been confirmed in two papers (Lally, Ghoshal & Fuchs, 2019; Gross et al., 2019).

Table 1 The performance of five algorithms on the training set.

AUPRC, area under the precision-recall curve. AUC, area under the receiver operating characteristic curve. AUPRC is the main metric. Sen, sensitivity. Spe, specificity. MCC, Matthews correlation coefficient. F1, F1-score. The selected algorithm and The maximum values in each metric are marked in bold.

Method	AUPRC	AUC	Sen	Spe	MCC	F1	
XGBoost	0.85	0.87	0.68	0.90	0.62	0.74	
SVM	0.84	0.86	0.68	0.85	0.55	0.70	
RF	0.82	0.86	0.70	0.85	0.56	0.72	
DT	0.80	0.82	0.68	0.84	0.53	0.70	
KNN	0.78	0.79	0.53	0.94	0.51	0.62	

Figure 2 Performance comparison of the five algorithms on the training set.

(A) AUPRC curves of the five algorithms. (B) AUC curves of the five algorithms.

In summary, the XGBoost algorithm outperformed other algorithms on most metrics. The AUPRC value on the training set is also the highest, which can better evaluate the model’s ability to correctly select positive samples. This is in line with our purpose of constructing IdentPMP.

IdentPMP outperforms other method

From the above steps, we finally chose XGBoost algorithm to construct IdentPMP. For performance evaluation, we compared the performance of IdentPMP and MPFit (Khan & Kihara, 2016) on the independent test set. MPFit is a calculation tool constructed by Khan et al. to predicting moonlighting proteins, mainly in the species of microorganisms and animals. This tool uses a variety of features, including Phylo (phylogenetic profiles), genetic interactions (GI), GE (gene expression profiles), DOR (disordered protein regions), NET (protein’s graph properties in the PPI network), PPI network. The two feature combinations MPFit (Phylo+DOR+NET+GE+GI) and MPFit (Phylo+PPI+GE) are recommended by the author. Then we used these two combinations to perform experiments on plant proteins (Khan, Mcgraw & Kihara, 2017).

In order to illustrate and compare the performance of IdentPMP with MPFit, we plotted the results of independent test set in Fig. 3. As we can see, the result shows that the AUPRC and the AUC of IdentPMP is 0.43 and 0.68, which are 19.44% (0.43 vs. 0.36) and 13.33% (0.68 vs. 0.6) higher than others, respectively. Other evaluation metrics can be seen in Table 2, IdentPMP is significantly better than any feature combination of MPFit. The accuracy of positive and negative samples for the MPFit (Phylo+DOR+NET+GE+GI) was 54% and 64%, respectively. When the MPFit (Phylo+PPI+GE) is executed, all samples are predicted to be positive samples. The physiological characteristics of plant and other species are quite different, and there is no plant data in the training set of other tools, which may be the reason for the low accuracy of other tools in predicting plant protein. IdentPMP used a variety of plant data and the TPC determined by plant characteristics to make the prediction of plant proteins perform well.

Figure 3 Performance comparison of IdentPMP and MPFit on the independent test set.

(A) AUPRC curves of the IdentPMP and MPFit (Phylo+DOR+NET+GE+GI). (B) AUC curves of the IdentPMP and MPFit (Phylo+DOR+NET+GE+GI).

Table 2 The detailed values of the results of IdentPMP and MPFit on the independent test set.

AUPRC, area under the precision-recall curve. AUC, area under the receiver operating characteristic curve. Sen, sensitivity. Spe, specificity. MCC, Matthews correlation coefficient. F1, F1-score. The maximum values in each metric are marked in bold.

Method	AUPRC	AUC	Sen	Spe	MCC	F1	
IdentPMP	0.43	0.68	0.46	0.89	0.37	0.52	
MPFit(Phylo+GE+GI+DOR+NET)	0.36	0.60	0.54	0.64	0.17	0.44	
MPFit(PPI+Phylo+GE)	0.64	0.50	1.00	0.00	0.00	0.43	

In summary, the observed results suggest that IdentPMP is a better and easier to use predictor specifically designed for plants. It can also be proved that a clearly defined benchmark data set containing both positive and negative samples is needed for the research on plant moonlighting proteins.

Conclusions

In this work, we constructed a benchmark data set and utilized feature class (TPC) to identify plant moonlighting protein. Then, we used sequence-based learning models to build a web-based prediction tool, IdentPMP. It is an integrated open-source tool for predicting the moonlighting proteins derived from plant species. As far as we know, the existing moonlighting protein prediction tools mainly focus on the proteins in animals and microorganisms. The IdentPMP is the first attempt to build a moonlighting protein prediction tool specific for plants. And it can be seen from the IdentPMP, and other tool’s prediction results that the prediction tool proposed here has a better performance.

Although IdentPMP performs well, there is still room for improvement. In future experiments, we will expand the data set to discover more plant moonlighting proteins using text mining, biological experiments and other methods. Using more advanced or powerful algorithms to build the model to improve the performance of prediction tools. At present, IdentPMP and the benchmark data set have filled the research gap of plant moonlighting protein. Its design principles and strategies can inspire bioinformatics to develop ideas for improved methods and can be applied tox other research topics in moonlighting protein analysis. We hope that IdentPMP will bring some benefits to the research field of plant moonlighting proteins.

Supplemental Information

Supplemental Information 1 The meaning of the feature classes and the number in each pretreatment step

Dimension, the original dimension of feature classes. IG (Information Gain), the feature dimension after feature selection using IG. PCA (Principal Comp onent Analysis), the feature dimension after dimensionality reduction using PCA method.

Click here for additional data file.

Supplemental Information 2 Modeling performance of each feature class (70%)

AUPRC, area under the precision–recall curve; AUC, area under the receiver operating characteristic curve. Sen, sensitivity. Spe, specificity. MCC, Matthews correlation coefficient. F1, F1-score. For those feature classes whose information entropy of all features is less than 0.05, the dimension of feature selection is set to 70%. The maximum values in each metric are marked in bold.

Click here for additional data file.

Supplemental Information 3 Modeling performance of each feature class

AUPRC, area under the precision–recall curve; AUC, area under the receiver operating characteristic curve. Spe, specificity. MCC, Matthews correlation coefficient. F1, F1-score. For those feature classes whose information entropy of all features is less than 0.05, the dimension of feature selection is set to 80%. The maximum values in each metric are marked in bold.

Click here for additional data file.

Supplemental Information 4 Modeling performance of each feature class (90%)

AUPRC, area under the precision–recall curve; AUC, area under the receiver operating characteristic curve. Sen, sensitivity. Spe, specificity. MCC, Matthews correlation coefficient. F1, F1-score. For those feature classes whose information entropy of all features is less than 0.05, the dimension of feature selection is set to 90%. The maximum values in each metric are marked in bold.

Click here for additional data file.

Supplemental Information 5 The performance of five algorithms on independent test sets

AUPRC, area under the precision–recall curve; AUC, area under the receiver operating characteristic curve. Sen, sensitivity. Spe, specificity. MCC, Matthews correlation coefficient. F1, F1-score. The maximum values in each evaluation metric are marked in bold.

Click here for additional data file.

Additional Information and Declarations

Competing Interests

Author Contributions

Data Availability

The authors declare there are no competing interests.

Xinyi Liu performed the experiments, analyzed the data, prepared figures and/or tables, designed and constructed website, and approved the final draft.

Yueyue Shen performed the experiments, prepared figures and/or tables, and approved the final draft.

Youhua Zhang and Fei Liu analyzed the data, authored or reviewed drafts of the paper, and approved the final draft.

Zhiyu Ma performed the experiments, prepared figures and/or tables, designed and constructed website, and approved the final draft.

Zhenyu Yue conceived and designed the experiments, analyzed the data, authored or reviewed drafts of the paper, and approved the final draft.

Yi Yue conceived and designed the experiments, authored or reviewed drafts of the paper, and approved the final draft.

The following information was supplied regarding data availability:

The code and data (UniProt ID of training set and PlantMP dataset) are available at http://identpmp.aielab.net/.

The Uniprot ID is used to query proteins in Uniprot.

https://www.uniprot.org/.

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
