# Peer review of "IdentPMP: identification of moonlighting proteins in plants using sequence-based learning models"

_PeerJ, doi:10.7717/peerj.11900_

## Round 0.1 · original submission · Major Revisions

Dear Dr. Yue,
Your manuscript has been evaluated by three independent reviewers, and all have pointed to several deficiencies in the current version.

Sincerely,

Gunjan

Reviewer 1 ·

Basic reporting

1. The manuscript needs to be checked for sentence construction and grammatical errors.
2. PMPIdent and IdentPMP has been interchangeably used in the figures and text.
3. Authors should make it more elaborate on how existing prediction tools (non-specific to plants) do not provide a suitable alternative for the IdentPMP.

Experimental design

1. Research methodology is well elaborated and appropriate statistical methods have been employed in the study.
2. Is the prioritized feature combination the same for both the tools, IdentPMP, and MPFit.
3. It is appreciative that IdentPMP is the first prediction tool for the plant moonlighting proteins. As the authors state in the results and discussion that the existing variability between IdentPMP and MPFit is perhaps due to the dominance of proteins from animals and microorganisms in the case of MPFit, shouldn't IdentPMP be compared with some of the existing databases for more confidence in the data?

Validity of the findings

1. How does the sensitivity and specificity curve comparison hold true between the two databases, IdentPMP and MPFit if the latter is predominant for animal proteins. Authors should mention in the text if the algorithm used to call for moonlighting function in the case of MPFit can be biased towards animal or plant proteins?
2. Authors should provide some information on comprehending the results obtained from the web interface. The user is provided with the score for each protein however, it will be beneficial to future users if information about the possible moonlighting roles can be integrated with the existing results.

Additional comments

Liu et al in their manuscript "IdentPMP: moonlighting proteins identification of plant using
sequence-based learning models" describes the first moonlighting function prediction tool for plant proteins. It is an important study that will not only shed light on the discovery of new moonlighting proteins but investigation into their functions will provide insights into their role in different cellular and organellar niches. The manuscript is suitable for publication provided the following comments are addressed by the authors.

Reviewer 2 ·

Basic reporting

.

Experimental design

.

Validity of the findings

.

Additional comments

The paper designed a sequence-based model to identify moonlighting proteins. Although moonlighting protein plays several functions, the work presented in this paper is still far from satisfactory. I do not think the paper is worth publishing based on following comments.
1. The number of samples, especially for positive samples is too small to have statistical significance.
2. Redundant sequences were not removed from benchmark data. Thus, their proposed model is overestimated.
3. Although authors used ilearn to generate features, the description about features are very poor. Readers cannot repeat their work. Moreover, how many features were used?
4. Authors did not perform enough feature analysis to show why these features are validated.
5. No webserver or softpackage was established.

Reviewer 3 ·

Basic reporting

The basic language is easy to follow however, the manuscript requires a serious revision to rectify errors related to the English language. I made some remarks in the abstract section below.

Line 22 “Hence, availability” should be “Hence, the availability”
Line 24 “data set that constructed by” should be “data set constructed by”
Line 25 “At the beginning” should be “in the beginning”.
Line 26 “plant protein,” incorrect use of punctuation “,”.
Line 37 “prediction tool were” should be “prediction tool was”
Line 29 & 30 “On the basis of again, ”, What does the author want to say is not clear in this sentence.


Text is provided with sufficient field background and context. Article structure and other components such as figure table etc. look fine.

Experimental design

1. The author included four ML algorithms to build models however there are several supervised methods such as decision trees, gradient boosting algorithms, etc were not included. The author did not provide any reasonable context to include these four and ignoring other algorithms.

2. MCC score produces a more informative and truthful score in evaluating binary classifications than accuracy. The author did not include MCC and F1 scores in this study to assess the models.

3. In the present time, advanced algorithms such as recurrent neural network and other deep learning methods combined with Natural language processing based algorithms and features such as word2vector outperforming conventional machine learning models to classify protein and peptide sequences. However, the author did not use any advanced neural network-based method to compare the results.

Validity of the findings

no comment

---

## Round 0.2 · Major Revisions

Your manuscript has been evaluated by two reviewers. One of the reviewers pointed out several deficiencies in the current version.

Reviewer 3 ·

Basic reporting

The revised manuscript has greatly improved, addressing all the comments provided based on the clarity, unambiguous and professional English.

Experimental design

The revised manuscript has addressed all the concerns and greatly improved.

Validity of the findings

The revised manuscript has addressed all the concerns and greatly improved.

Reviewer 4 ·

Basic reporting

Authors Liu et al developed IdentPMP- an SVM based predictor as a webserver, for predicting plant moonlighting proteins. The authors tried five different classification methods of which they claim to discover SVM to be working best.

The major strength is the right target of problem statement, that there are no plant moonlighting protein predictors. IdentPMP, being the first of its kind has merit and may benefit the plant community.

The limitation is of unbalanced data size for training and testing, with low number of training and test cases. The metrics of evaluation also do not show particularly great results.

Experimental design

1. Authors used several approaches but decided SVM is the best, while table 1 does not really support this inference. I think nearly all metrics being similar, overall RF seems to be the best. Deciding upon SVM based on a closely matching AUPRC -which again is a strange metric for evaluation instead of AUROC, unless the base positive ratios is known/clearly mentioned. To evaluate AUPRC, isn’t normal for all types of readers and authors did not mention it for clarity. There isn’t much to choose between RF and SVM. It appears that SVM was used and other methods were thrown into the mix as an afterthought.
2. After the predictor was made, I expected some metrics to be shown for training set as well as test set. Then followed by a prediction to show its use case for some plant proteome, to demonstrate new findings, which is missing.
3. Did not understand the feature engineering section as it is filled with too much jargon, at least for me. Too many abbreviations without context. The users may not be SVM researchers.
4. Why were 80% features chosen? Rationale of many such choices isn’t very clear. Another is AUPRC, no cutoff description means users cannot decide what is good and why. AUPRC is not straight forward like AUC, why was it chosen if the rationale and interpretation cannot be given.

Validity of the findings

1. A gain of 13.89% in AUPRC and 10% in AUROC needs to be explained in context, over other methods as 0.41 (AUPRC) and 0.66 (AUROC) appear to be quite low values. This is not clear how these % improvements numbers are calculated. Also, the increase of 10% to get 0.66 may denote the previous best was 0.56 (?) for which other predictor (which is not plant moonlighting one) was used. It needs to be explained in better context because a method nto suited is bound to give such results (but 0.56 for a generic predictor is still bad and unexpected). How does MPFit perform for its own use case scenario?
2. The paper requires a careful editing to correct for weak reasoning and arguments. It does not help the reader when they have to assume and fill in the interpretations and reasons behind the work, or interpretation of data properly. Flow of text from one argument to another seems to be missing.

Additional comments

1. How much time does the tool take to run? Mandating email address for every single sequence may not be prudent (even for example sequence?). As reviewer, I could not use the tool due to this reason.
2. The language is a challenge to read with lots of grammatical mistakes. A careful editing is needed. Some examples are -
i. Line 23-24, change “by ourselves” to “in house”
ii. Line 45, change “overturned,” to “overturned by”
iii. Line 52, delete “species’”
iv. Line 55, perhaps authors meant “genomes” instead of “genes”
v. Line 57, delete “very” , change “important,” to “important that”
vi. Line 58, change “some” to “effective”
vii. Line 59, change “some” to “the”
viii. Line 70, delete “ tools regarding”
ix. Line 71, delete “prediction tools to identify moonlighting proteins”
x. Line 77, change “mainly identifying” to “ mainly to identify”
xi. Line 78, change “organisms,” to “organisms, but” AND change “Considering that” to “ The”
xii. Line 79, “microbial” to “microbes”

I gave up after this and may have certainly missed many places.
3. Line 119, 2/3 rd data for each species is practically impossible for many species with 4, 2, 2, proteins.
4. Features and feature class needs to be clearly defined, are they same?
5. Sudden use of abbreviations at line 236?
6. Results discussion /Conclusion needs proper thoughtful writing. Nothing is obvious what is claimed. 0.66 AUROC isn’t a great result.

---

## Round 0.3 · accepted · Accept

The authors have satisfactorily answered all of the reviewers' concerns.

Reviewer 4 ·

Basic reporting

The manuscript has made necessary changes to improve the content, the flow has improved a lot and the errors have been corrected.

Experimental design

The revised manuscript has addressed the concerns and is improved a lot.

Validity of the findings

The revised manuscript has addressed the concerns and is improved a lot.

Additional comments

No further comments